# CAB: Comprehensive Attention Benchmarking on Long Sequence Modeling

## Abstract

Transformer has achieved remarkable success in language, image, and speech processing. Recently, various efficient attention architectures have been proposed to improve transformer's efficiency while largely preserving its efficacy, especially in modeling long sequences. A widely-used benchmark to test these efficient methods' capability on long-range modeling is Long Range Arena (`LRA`). However, `LRA` only focuses on the standard bidirectional (or noncausal) self attention, and completely ignores cross attentions and unidirectional (or causal) attentions, which are equally important to downstream applications. Although designing cross and causal variants of an attention method is straightforward for vanilla attention, it is often challenging for efficient attentions with subquadratic time and memory complexity. In this paper, we propose Comprehensive Attention Benchmark (`CAB`) under a fine-grained attention taxonomy with four distinguishable attention patterns, namely, noncausal self, causal self, noncausal cross, and causal cross attentions. `CAB` collects seven real-world tasks from different research areas to evaluate efficient attentions under the four attention patterns. Among these tasks, `CAB` validates efficient attentions in eight backbone networks to show their generalization across neural architectures. We conduct exhaustive experiments to benchmark the performances of nine widely-used efficient attention architectures designed with different philosophies on `CAB`. Extensive experimental results also shed light on the fundamental problems of efficient attentions, such as efficiency length against vanilla attention, performance consistency across attention patterns, the benefit of attention mechanisms, and interpolation/extrapolation on long-context language modeling.

## 1 Introduction

Transformer has achieved great breakthroughs in natural language processing (Devlin et al., 2019; Radford et al., 2019; Brown et al., 2020), computer vision (Dosovitskiy et al., 2020; Liu et al., 2021b), speech processing (Schneider et al., 2019; Ren et al., 2021), and biology (Jumper et al., 2021; Brandes et al., 2022). A major drawback of the transformer architecture is its quadratic complexity in both time and memory. The problem has been more evident with the ever-increasing need in applying transformers for longer sequence modeling in different domains.

Recent research on efficient attention mechanisms seeks to respond to this problem by improving attention efficiency while preserving efficacy (Wang et al., 2020; Kitaev et al., 2019; Zaheer et al., 2020; Choromanski et al., 2020; Zheng et al., 2022). The commonly adopted test bed for benchmarking efficient transformers in the context of processing long sequences, is the Long Range Arena (Tay et al., 2020a; `LRA`), consisting of both synthetic probing tasks and real-world tasks. However, all of these tasks focus on the *self attention* setting, ignoring *cross attention* and *causal attention*, which are equally important and often more challenging. In other words, the transformer model is only used as a sequence encoder in `LRA`, while in real applications, *cross attention* is essential for conditionality modeling tasks such as sequence-to-sequence (Bahdanau et al., 2015), data-to-text (Dušek et al., 2020) and knowledge-enhanced models (Liu et al., 2021a), and *causal attention* is critical for causality modeling tasks such as text generation (Vaswani et al., 2017; Zhang et al., 2020), language modeling (Radford et al., 2019; Brown et al., 2020) and speech synthesis (Li et al., 2019). Another potential drawback of `LRA` as discovered by researchers recently, is that with proper tuning, the performance gap between different transformer variants in these tasks can be insignificant (Xiong et al., 2022; Ivgi et al., 2022; Shaham et al., 2022), impairing its effectiveness as a standard benchmark.

To address these problems, we propose a comprehensive attention benchmark (CAB) for long sequence modeling. Above all, we present a fine-grained attention taxonomy, considering attentive functionality on conditionality and causality modeling. We present four patterns of attentions, namely, noncausal self, causal self, noncausal cross, and causal cross, representing the distinguishable attentive functionality to sequence modeling (§2). With that in mind, we then collect seven real-world tasks from diverse fields of computer vision, natural language processing, speech processing, and time series forecasting (§3). Among these tasks, CAB includes rich backbone architectures to evaluate the attention mechanisms, testing their performances and generalization abilities. Given four attention patterns defined by the attention taxonomy, we advocate a pattern-wise comparison between attention mechanisms, evaluating their attentive functionality respectively.

We conduct exhaustive experiments on CAB, assessing nine widely-used efficient attentions designed with different philosophies (§4). The experimental results reveal several insights into designing efficient attention architectures. First, we show that existing efficient transformers claiming comparable or even superior performances to vanilla attention, often achieve less competitive results in the causal cross scenario, indicating efficient attentions' modeling capability cannot always generalize across different attention patterns. Second, by quantifying the efficiency of attentions in long sequence contexts using *efficiency length* (i.e., the minimum length for a sub-quadratic efficient model to surpass vanilla attention in efficiency), we disclose the underlying inefficiency problem of existing efficient attention methods in modeling relatively short sequences. Third, we investigate interpolation and extrapolation on a long-context language model, and find that it is promising for efficient attention, such as local attention and LongShort Transformer, to scale to long-context language modeling. We hope CAB and elaborated experimental efforts can shed light on the fundamental problems of efficient attentions, and inspire the design of advanced attention mechanisms. CAB and all the related codes will be released at https://github.com/Anonymous.

## 2 ATTENTION TAXONOMY

Attention methods are designed to capture token-to-token dependency within and across sequences. Let $Y = \{y_1, y_2, \ldots, y_n\}$ and $\mathbf{Y} \in \mathbb{R}^{n \times d}$ be a target sequence and its feature matrix, and $X = \{x_1, x_2, \ldots, x_m\}$ and $\mathbf{X} \in \mathbb{R}^{m \times d}$ be a source sequence and its feature matrix, where $n$ and $m$ are target and source sequence lengths respectively, and $d$ denotes dimensionality. Note that $\mathbf{X}$ could be the same as $\mathbf{Y}$ in the case of self attention. The target and source feature matrices are transformed to **Q**uery-**K**ey-**V**alue feature matrices as $\mathbf{Q} = YW_{\mathbf{Q}}, \mathbf{K} = XW_{\mathbf{K}}, \mathbf{V} = XW_{\mathbf{V}}$ with learnt parametric matrices $W_{\mathbf{Q}}, W_{\mathbf{K}}, W_{\mathbf{V}}$. Vanilla transformer (Vaswani et al., 2017) learns to integrate key-value feature matrices $\mathbf{K} \in \mathbb{R}^{m \times d}, \mathbf{V} \in \mathbb{R}^{m \times d}$ into a query one $\mathbf{Q} = \{\mathbf{q}_1, \mathbf{q}_2, \ldots, \mathbf{q}_n\} \in \mathbb{R}^{n \times d}$ with token-to-token attention:

$$\text{Attn}(\mathbf{Q}, \mathbf{K}, \mathbf{V}) = \text{softmax}\left(\mathbf{Q}\mathbf{K}^\top / \sqrt{d}\right) \mathbf{V} \tag{1}$$

in which we omit the multihead notation without loss of generality. Vanilla attention in Eq. 1 advocates a token-to-token alignment matrix $\mathbf{Q}\mathbf{K}^\top$, which results in quadratic complexity $O(nm)$ of computational time and memory usage. It is observed that $\text{Attn}(\cdot)$ in Eq. 1 acts as an integration model, projecting query features $\mathbf{Q}$ by integrating key-value features without changing $\mathbf{Q}$'s shape. Beyond the vanilla token-to-token attention mechanism, we extend Eq. 1 to a more general form of attention family as:

$$\text{Attn}(\mathbf{Q}, \mathbf{K}, \mathbf{V}) = f(\mathbf{Q}; \mathbf{K}, \mathbf{V}) : \mathbb{R}^{n \times d} \to \mathbb{R}^{n \times d}$$
$$\text{where } \text{Attn}_i(\mathbf{Q}, \mathbf{K}, \mathbf{V}) = f_i(\mathbf{q}_i; \mathbf{Q}, \mathbf{K}, \mathbf{V}), i = 1 \ldots n \tag{2}$$

where $\mathbf{Q}$ is treated as the main variable and $\mathbf{K}, \mathbf{V}$ are conditional ones. Comparing with Eq. 1, Eq. 2 covers efficient attention methods without explicitly modeling the token-to-token alignment. In usage, attentions are modified to fit generative models with underlying structures, such as conditional models and causal models, to achieve specific attentive functionality.

In this section, we put forward a fine-grained attention taxonomy, considering the conditionality and causality of attentions. We show their challenges and potential impact on real-world applications in modeling conditionality and causality. Under the taxonomy, we present four attention patterns with different attentive functionality as shown in Figure 1.

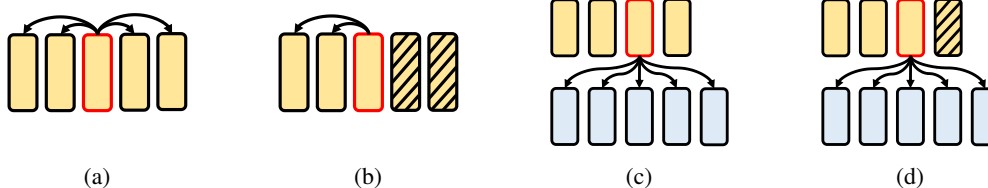

Figure 1: Computation diagrams for (a) noncausal self, (b) causal self, (c) noncausal cross, (d) causal cross attentions. Shaded blocks represent future tokens that are invisible to the current state. Blocks with red rims represent the current state token.

**Self Attention & Cross Attention** Self attention (Lin et al., 2017; Vaswani et al., 2017; Dosovitskiy et al., 2020) learns contextual features within a sequence whereas cross attention (Bahdanau et al., 2015) learns to integrate features across sequences. Both self attention and cross attention are highly effective in sequence learning, but designing cross attention is more challenging for efficient attentions without explicit token-to-token alignment. On the aspect of computation, $\mathbf{Q} \in \mathbb{R}^{n \times d}$ and $\mathbf{K}, \mathbf{V} \in \mathbb{R}^{m \times d}$ are of different shapes and are impossible to perform elementwise operations along length dimension, such as cross-covariance used in XCiT (Ali et al., 2021). Besides, different from self attention, cross attention lacks implicit alignment between $\mathbf{Q}$ and $\mathbf{K}, \mathbf{V}$, such as neighborhood information. Cross attention learns to integrate conditional features $\mathbf{K}, \mathbf{V}$ into the main features $\mathbf{Q}$, and has been successfully applied to real-world applications, such as machine translation (Bahdanau et al., 2015; Vaswani et al., 2017), data-to-text generation (Dušek et al., 2020; Shen et al., 2020) and knowledge enhanced model (Jin et al., 2020; Liu et al., 2021a).

**Noncausal Attention & Causal Attention** While modeling a sequence, noncausal models have a holistic view of the whole sequence whereas causal ones only access to previous tokens up to now.[1] This restriction forbids efficient attention like Nyströmformer (Xiong et al., 2021), which must be conditioned on the whole target sequence to achieve linearity, to be applied in causal models. Furthermore, when training a causal model, efficient attentions such as RFA (Peng et al., 2021) and ABC (Peng et al., 2022) have to provide different sets of features from $\mathbf{Y}$ for each query token. This mechanism in these attentions, therefore, leads to multiplied memory consumption compared to their noncausal counterparts. We provide a detailed analysis of the increasing computation costs in Appendix B. In real-world applications, causal attentions bring up strong generation capability and is widely adopted in text generation (Bahdanau et al., 2015; Vaswani et al., 2017; Zhang et al., 2020), text-to-speech synthesis (Li et al., 2019) and language modeling (Radford et al., 2019; Brown et al., 2020; Raffel et al., 2020; Chen et al., 2021a).

**Attention Patterns** As discussed above, conditionality and causality are two orthogonal aspects of modeling sequences. Thus we obtain four attention patterns via Cartesian product as $\{\text{noncausal}, \text{causal}\} \times \{\text{self}, \text{cross}\}$: noncausal self (NS), causal self (CS), noncausal cross (NC), and causal cross (CC). Following the general form of attentions in Eq. 2, we formulate the four attention patterns as below:

$$\text{NS}_i(\mathbf{Q}, \mathbf{K}, \mathbf{V}) \triangleq f_i(\boldsymbol{q}_i; \mathbf{Q}, \mathbf{K}, \mathbf{V}), s.t.\ \mathbf{Y} = \mathbf{X} \tag{3}$$

$$\text{CS}_i(\mathbf{Q}, \mathbf{K}, \mathbf{V}) \triangleq f_i(\boldsymbol{q}_i; \mathbf{Q}_{\leq i}, \mathbf{K}_{\leq i}, \mathbf{V}_{\leq i}), s.t.\ \mathbf{Y} = \mathbf{X} \tag{4}$$

$$\text{NC}_i(\mathbf{Q}, \mathbf{K}, \mathbf{V}) \triangleq f_i(\boldsymbol{q}_i; \mathbf{Q}, \mathbf{K}, \mathbf{V}) \tag{5}$$

$$\text{CC}_i(\mathbf{Q}, \mathbf{K}, \mathbf{V}) \triangleq f_i(\boldsymbol{q}_i; \mathbf{Q}_{\leq i}, \mathbf{K}, \mathbf{V}) \tag{6}$$

Figure 1 illustrates the computation diagrams of these four patterns. These attention patterns construct typical sequence models with attention mechanisms. For instance, Transformer (Vaswani et al., 2017) adopt NS attention in encoder, and CS/CC ones in decoder; recently prevailing Non-Autoregressive Transformer (Gu et al., 2018) models adopt NS attention in the encoder, and NS/NC attention in the decoder. Previous efficient attention benchmark LRA only tests attentions with the NS pattern but

---

[1]When generating a sequence $Y = \{y_1, y_2, \ldots, y_n\}$ autoregressively, neural models (Vaswani et al., 2017; Li et al., 2019; Brown et al., 2020) usually adopt a causal graph $\mathcal{G}(V, E)$ with vertices $V = \{y_i \in Y\}$ and edges $E = \{y_i \rightarrow y_j | i \in [1, \ldots, n], j \in [1, \ldots, n], i \leq j\}$.

Table 1: Task statistics of datasets, data lengths, evaluation metrics, backbone neural networks and required attention patterns. For encoder-decoder models, both source and target lengths are listed.

| Task | Dataset | Length | Metric | Model | Noncausal Self | Causal Self | Noncausal Cross | Causal Cross |
|------|---------|--------|--------|-------|:---:|:---:|:---:|:---:|
| TTS | LJSpeech | 559 | MCD MSD | FastSpeech 2 | ✔ | ✗ | ✗ | ✗ |
| | | 100/141 | | Transformer-TTS | ✔ | ✔ | ✗ | ✔ |
| Sum | Multi-News | 4,096/400 | ROUGE | Transformer | ✔ | ✔ | ✗ | ✔ |
| LSTF | ETT | 336/720 | MSE MAE | Informer | ✔ | ✗ | ✔ | ✗ |
| | ECL | 336/960 | | | | | | |
| | Weather | 720/720 | | | | | | |
| PCC | PCN | 1,536/3,584 | CD F-Score | PoinTr | ✔ | ✗ | ✔ | ✗ |
| LM | PG-19 | 8,192 | PPL | GPT-2 | ✗ | ✔ | ✗ | ✗ |
| MLM | | 2,048 | | RoBERTa-base | ✔ | ✗ | ✗ | ✗ |
| SR | FFHQ CelebA-HQ | 16,384 | PSNR SSMI | SR3 | ✔ | ✗ | ✗ | ✗ |

without the others, leading to an insufficient evaluation of attentive functionality. In the subsequent sections, we will introduce a comprehensive attention benchmark CAB that evaluates attentions comprehensively on the four patterns.

## 3 COMPREHENSIVE ATTENTION BENCHMARK

We propose Comprehensive Attention Benchmark (CAB) that collects seven tasks covering four research fields, including computer vision, natural language processing, speech processing, and time series forecasting. We introduce these tasks with their datasets, data lengths, evaluation metrics, and backbone neural networks. Then we present a compositional index (*CI*) to combine individual task performance into a compositional one for direct performance comparison across tasks and models.

### 3.1 LONG SEQUENCE MODELING TASKS

We collect seven real-world long-sequence modeling tasks in CAB with data lengths from 300 to 16,000 and include eight widely-used models as backbones in CAB to assess typical efficient attention mechanisms. Table 1 summarizes the tasks' statistics, evaluation metrics, backbone neural networks and the required attention patterns (§2) are also checked.

**Text-to-Speech Synthesis (TTS)**  This task requires models to convert input text sequences, descriptions, or narrations produced by a single speaker, to synthesized audio clips sharing the same timbre as the provided speaker. The CAB incorporates the LJSpeech dataset (Ito, 2017) whose audio clips are sampled with 22,050 Hz. Under this set of relatively high sample rates, the average sequence length of processed audio clips is 559. We use non-autoregressive FastSpeech 2 (Ren et al., 2021) and autoregressive Transformer-TTS (Li et al., 2019) as backbone networks. We adopt Mean Cepstral Distortion (MCD) and Mel Spectral Distortion (MSD) for objective evaluation.

**Summarization (Sum)**  This task is one that makes a comprehensive summary of multiple input documents without losing important information. We consider multi-document summarization task and use Multi-News datasets (Fabbri et al., 2019) for long sequence modeling. To make the task more challenging, we set the maximum source text and target summary lengths to be 4,096 and 400, respectively. Transformer (Vaswani et al., 2017) implemented on PARAGEN (Feng et al., 2022) serves the evaluating backbone and ROUGE (R-N) (Lin, 2004) is used to evaluate each attention.

**Long Sequence Time-series Forecasting (LSTF)**  Long sequence time-series forecasting is to predict long-term future behavior based on past states. This task evaluates models on three datasets,

including Electricity Transformer Temperature (ETT), Electricity Consuming Load (ECL), and Weather (Zhou et al., 2021). Following (Zhou et al., 2021), we use Informer as backbone and conduct univariate and multivariate evaluations. We average their Mean Square Error (MSE) and Mean Absolute Error (MAE) to obtain the final scores.

**Point Cloud Completion (PCC)**    This task is to complete a frame of point cloud data when providing partial points. `CAB` adopts PCN dataset (Griffiths & Boehm, 2019) which inputs 2,048 3D point coordinates and requires models to output the complete point cloud data comprising 14,336 points. PoinTr (Yu et al., 2021) is adopted here serving as evaluating backbone. Chamfer Distance (CD) (Huang et al., 2020) and F-Score (Tatarchenko et al., 2019) are selected to evaluate each attention. For clearer comparison, CD metrics are enlarged by 1,000 times.

**Language Modeling (LM)**    The language modeling task requires the language model to predict the next text based on the previous information. In this task, we consider a long-context language modeling where the context length is prolonged to 8,192. We use PG-19 dataset (Rae et al., 2019) and conduct evaluations based on the backbone model GPT-2 (Radford et al., 2019) implemented on FAIRSEQ (Ott et al., 2019). Perplexity (PPL) is used to serve as the evaluation metric.

**Masked Language Modeling (MLM)**    Different from the language modeling task, masked language modeling (Devlin et al., 2019) uses full context to predict masked tokens. Roberta-base (Liu et al., 2019) implemented on FAIRSEQ is adopted as the evaluation backbone. We use the same dataset and evaluation metric as LM task while the context length is set to 2,048.

**Super-Resolution (SR)**    In this task, we aims to convert low-resolution ($16 \times 16$) face images into high-resolution ($128 \times 128$) images. Following Saharia et al. (2022), we train the backbone model SR3 on Flickr-Faces-HQ (FFHQ) dataset (Karras et al., 2019) and conduct evaluation on CelebA-HQ dataset (Karras et al., 2018). Peak Signal-to-Noise Ratio (PSNR) and Structural SIMilarity (SSIM) (Wang et al., 2004) are selected to evaluate models from different aspects.

To test attentions under different patterns, we assign each pattern with tasks as suggested in Table 1. We exclude LSTF and PCC tasks from noncausal self attention test for the limited contribution of attentions on them (see §4.2).

## 3.2 Compositional Indexes

We compute a compositional index (*CI*) from seven tasks for performance comparison. CI is a normalized score to balance the influence among evaluation metrics, and high CI represents excellence. It is computed as follows: a) we transform all evaluation metrics beforehand, so that a higher score indicates better performance; b) we then normalize each transformed metric with Z-score normalization; c) after normalization, the score of each evaluation metric is averaged within each task, and is further averaged across tasks. We include details of CI calculation in Appendix C.

## 4 Experiments

### 4.1 Main Results

We conduct exhaustive experiments on `CAB` by applying various efficient attentions to the backbone models and measure their performance under noncausal self, causal self, noncausal cross, and causal cross attention patterns. We extensively study the fundamental problems of efficient attentions, in terms of efficiency length against vanilla attention, performance consistency across attention patterns, the benefit of attention, and interpolation/extrapolation on long-context language modeling. The details of experimental settings for tasks are shown in Appendix D.

**Compared Efficient Attentions**    We benchmark nine typical efficient attentions designed with different philosophies, including sparse attention (**local attention** (Luong et al., 2015)), kernel-based linearized attention (**Performer** (Choromanski et al., 2020), **cosFormer** (Qin et al., 2021), **LARA** (Zheng et al., 2022)), low-rank factorization (**Nyströmformer** (Xiong et al., 2021)), pro-

Table 2: Efficient attentions with supported attention patterns. The complexity is based on sequence length $n$, omitting independent factors for simplicity. ($*$) denotes the attentions that perform differently as the original paper claims. We clarify the reasons in Appendix A.

| Method | Attention Name | Complexity | Noncausal Self | Causal Self | Noncausal Cross | Causal Cross |
|---|---|---|---|---|---|---|
| *Sparsity* | local attention | $O(n)$ | ✔ | ✔ | ✗ | ✗ |
| *Kernel* | LARA | $O(n)$ | ✔ | ✗ | ✗ | ✗ |
| | cosFormer $*$ | $O(n)$ | ✔ | ✗ | ✔ | ✗ |
| | Performer $*$ | $O(n)$ | ✔ | ✗ | ✔ | ✔ |
| *Low-rank factorization* | Nyströmformer | $O(n)$ | ✔ | ✗ | ✗ | ✗ |
| *Prototype* | ABC | $O(n)$ | ✔ | ✔ | ✔ | ✔ |
| | ProbSparse | $O(n \log n)$ | ✔ | ✗ | ✗ | ✗ |
| *Hybrid* | LongShort | $O(n)$ | ✔ | ✔ | ✗ | ✗ |
| *State space* | S4D | $O(n)$ | ✔ | ✔ | ✗ | ✗ |

totype (**ABC**[2] (Peng et al., 2022), **ProbSparse** (Zhou et al., 2021)), hybrid attention (**LongShort Transformer** (Zhu et al., 2021)), and state space (**S4D** (Gu et al., 2022)) and evaluate these attention architectures in terms of different attention patterns described in §2. We provide a brief overview of these attention architectures in Table 2 and leave a detailed description in Appendix F.

The main results are shown in Table 3. It is observed that efficient attention achieves comparable performance with vanilla attention under noncausal self attention pattern, and some of them even surpass vanilla attention. It suggests that the current efficient attention indeed improves the ability to learn contextual features in long sequences of various modalities. Although the performance of efficient attention is remarkable in the noncausal self pattern, it is less competitive in the other three patterns. As shown in Table 3d, causal cross attentions are extremely hard to design and are inferior to vanilla attention by a significant margin. Due to the challenge of causality modeling, it is notable that when compared with autoregressive models (i.e. Transformer and Transformer-TTS), non-autoregressive or parallel generation models (i.e. FastSpeech 2, SR3, and RoBERTa) are more likely to have their efficient variants without significant performance drops.

On the dimension of efficient attention, we find that local attention is a strongly competitive efficient model on most real-world tasks, which agrees with the previous studies (Xiong et al., 2022), although it is utterly defeated in LRA. It achieves the highest score of 0.978 under the noncausal self pattern and ranks second under the causal self pattern. In contrast, S4D, which achieves impressive results on LRA, appears less inspiring than expected on Sum and MLM tasks under the noncausal self attention pattern.

**Efficiency Analysis** We conduct a simulation efficiency experiment under noncausal self attention pattern. The experiment is performed on a single A100 GPU, where attention mechanisms are fed a set of dummy sequences with lengths of $\{256, 512, 1024, 2048, 4096, 8192\}$. Table 2 shows the time and memory consumption against sequence length by each attentions. We can find that the advantage of efficient attention gradually expands as the sequence length increases. Moreover, kernel-based and low-rank attentions are more efficient on time and space. Further, we report *efficiency length* to measure the utility of efficient attentions. The efficiency length is defined as the intersection point of computational time and memory curves by efficient models and vanilla attention. The efficiency length represents the minimum length that a sub-quadratic efficient model surpasses vanilla attention in efficiency. Specifically, we perform regression on vanilla and efficient attentions by their theoretical complexity. Then we calculate the intersection of the regression line and the quadratic regression curve to get an efficiency length. Figure 2c shows that efficient attentions gain superiority over vanilla attention once the sequence contains thousands of tokens. For noncausal self attention, ABC and Performer have the shortest efficiency lengths on running time and memory usage, respectively. The details of efficiency lengths and experiment settings can be found in Appendix G.

---

[2]We implement and experiment ABC with its fast version as suggested by Peng et al. (2022).

Table 3: Experimental results on `CAB` with (a) noncausal self, (b) causal self, (c) noncausal cross, and (d) causal cross attention pattern. (-) denotes that the method fails on the task. (†) denotes that the attention does not support fp16 training. The CI of each efficient attention is shown in Appendix E.

| Method | TTS | | | | Sum | | | SR | | MLM | CI↑ |
|---|---|---|---|---|---|---|---|---|---|---|---|
| | FastSpeech 2 | | Transformer-TTS | | Transformer | | | SR3 | | RoBERTa | |
| | MCD↓ | MSD↓ | MCD↓ | MSD↓ | R-1↑ | R-2↑ | R-L↑ | PSNR↑ | SSMI↑ | PPL↓ | |
| vanilla | 3.475 | 1.974 | 4.095 | 2.199 | 34.61 | 6.35 | 31.66 | 23.18 | 0.675 | 3.42 | -0.024 |
| local | 3.419 | 1.970 | 4.015 | 2.164 | **38.50** | **10.54** | **35.39** | 23.33 | 0.682 | 4.18 | **0.978** |
| cosFormer† | 3.400 | 1.956 | 4.030 | 2.160 | 34.77 | 6.34 | 31.74 | **23.53** | **0.690** | 5.25 | 0.466 |
| LongShort | 3.436 | 1.996 | **3.913** | **2.136** | 34.35 | 6.41 | 31.55 | 23.28 | 0.681 | **3.38** | 0.269 |
| LARA | 3.463 | 2.012 | 4.116 | 2.209 | 34.03 | 6.23 | 31.23 | 23.35 | 0.685 | 6.45 | -0.032 |
| Performer | 3.437 | 1.983 | 4.115 | 2.198 | 34.85 | 6.54 | 31.88 | 23.34 | 0.682 | 111.73 | -0.285 |
| Nyströmformer | 3.557 | 2.036 | 4.274 | 2.276 | 34.45 | 6.30 | 31.56 | 23.20 | 0.679 | 6.32 | -0.440 |
| ProbSparse | 3.363 | 1.946 | 4.034 | 2.161 | 34.62 | 6.36 | 31.64 | 22.98 | 0.667 | 186.35 | -0.661 |
| ABC | 3.393 | 1.966 | 4.085 | 2.204 | 33.80 | 6.07 | 30.98 | 22.54 | 0.635 | 10.50 | -0.686 |
| S4D† | **3.303** | **1.905** | 4.017 | 2.195 | | – | | 23.35 | 0.682 | 25.34 | – |

(a)

| Method | TTS | | Sum | | | LM | CI↑ |
|---|---|---|---|---|---|---|---|
| | Transformer-TTS | | Transformer | | | GPT-2 | |
| | MCD↓ | MSD↓ | R-1↑ | R-2↑ | R-L↑ | PPL↓ | |
| vanilla | 4.095 | 2.198 | 34.61 | 6.35 | 31.66 | – | – |
| S4D | **4.030** | **2.188** | **34.90** | **6.65** | **31.98** | 15.78 | **0.985** |
| LongShort | 4.039 | 2.195 | 33.55 | 6.27 | 30.71 | **15.52** | 0.617 |
| local | 4.141 | 2.220 | 33.50 | 6.27 | 30.74 | 19.73 | -0.239 |
| ABC | 4.058 | 2.189 | 30.17 | 5.48 | 27.92 | 29.13 | -0.623 |

(b)

| Method | PCC | | | LSTF | | | | | | CI↑ |
|---|---|---|---|---|---|---|---|---|---|---|
| | PoinTr | | | Informer | | | | | | |
| | CD-$\ell_1$↓ | CD-$\ell_2$↓ | F-Score↑ | ETT | | Weather | | ECL | | |
| | | | | MSE↓ | MAE↓ | MSE↓ | MAE↓ | MSE↓ | MAE↓ | |
| vanilla | 7.425 | 0.231 | 0.779 | **1.138** | **0.775** | 0.478 | **0.508** | **0.449** | **0.486** | **0.596** |
| ABC | **7.344** | **0.227** | **0.784** | 1.147 | 0.809 | 0.489 | 0.520 | 0.519 | 0.527 | 0.204 |
| Performer | 7.368 | 0.235 | 0.783 | 1.254 | 0.819 | **0.463** | **0.508** | 0.881 | 0.722 | 0.014 |
| cosFormer | 8.673 | 0.298 | 0.704 | 1.219 | 0.823 | 0.474 | 0.509 | 0.632 | 0.590 | -0.815 |

(c)

| Method | TTS | | Sum | | | CI↑ |
|---|---|---|---|---|---|---|
| | Transformer-TTS | | Transformer | | | |
| | MCD↓ | MSD↓ | R-1↑ | R-2↑ | R-L↑ | |
| vanilla | **4.095** | **2.198** | **34.61** | **6.35** | **31.66** | **0.956** |
| ABC | 5.780 | 2.631 | 32.22 | 5.55 | 29.53 | 0.058 |
| Performer | 6.635 | 3.053 | 27.22 | 3.88 | 25.21 | -1.014 |

(d)

## 4.2 DISCUSSIONS

**Performance Consistency Across Attention Patterns**  We investigate whether efficient attentions perform consistently across different attention patterns. To achieve this, we compute Pearson correlation (Benesty et al., 2009) over CI between each pair of attention patterns against the same arrangement of efficient attentions[3] for intra-benchmark comparison. Note that we also include CI computed from `LRA` for inter-benchmark comparison between `CAB`'s four attention patterns and `LRA`. In Figure 3a, `CAB`'s NS attention pattern is highly correlated with `LRA`, reflecting that efficient NS

---

[3]The order is [vanilla, local, ABC, LARA, cosFormer, ProbSparse, Performer, LongShort, Nyströmformer, S4D]

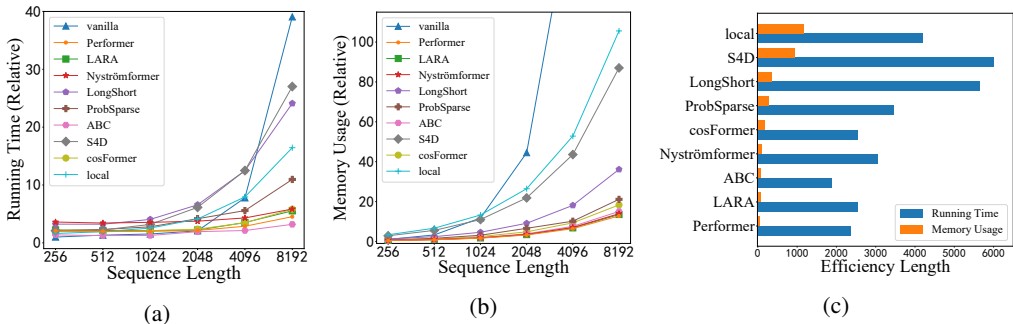

(a)                                  (b)                                  (c)

Figure 2: Empirical running time (a) and memory cost (b) with sequence length. Relative measurements to vanilla attention are reported. (c) efficiency length of attention architectures compared to vanilla attention. The efficient attention order on the y-axis is monotonically sorted by efficiency length on memeory usage.

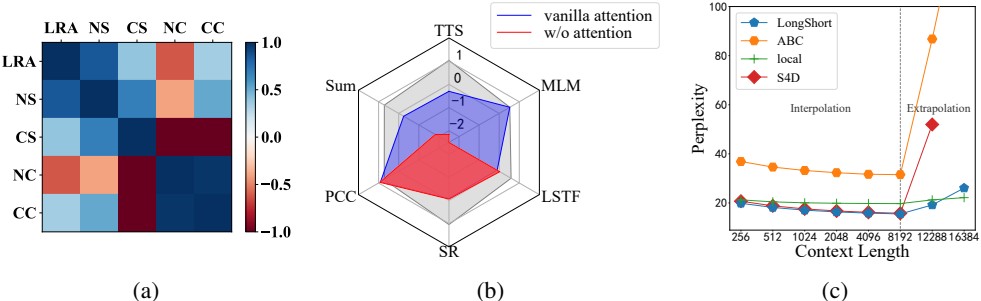

(a)                                  (b)                                  (c)

Figure 3: (a) Pairwise attention pattern correlation where NS, CS, NC, and CC signify noncausal self, causal self, noncausal cross, and causal cross attention respectively; (b) Ablation study of removing attention. The gray part is the score achieved by the existing efficient attention family, where we select the most well-performed efficient attention for each task; (c) the performance of efficient attentions based on different context lengths in the LM task during the test phase. S4D fails on the context with 16,384 tokens.

attentions developed on LRA do contribute to a wide range of real-world applications. In contrast, cross attention patterns share non-positive correlations with self attention patterns, indicating their inconsistent performance on self and cross attention.[4] On the causality dimension, the performances of causal and noncausal models are highly related but not strictly consistent. We also include task-to-task correlation analysis in Appendix I.

**Benefit of Attention** Self attention is designed to capture contextual features while cross attention integrates non-homologous information. But how much do these features benefit the model? We conduct an ablation study without the attention mechanism. In a preliminary experiment, we find that causal and cross patterns are indispensable to the backbone models as expected, while noncausal self attention only has a limited contribution. Therefore, we focus on discussing the attentive functionality of noncausal self attention. Figure 3b depicts the CI of backbone models with and without attention on tasks. Detailed CI can be found in Appendix H. Results show that attention mechanism improves performance on most tasks. Embarrassingly, after removing the attention mechanism, the PoinTr and Informer achieve improvement on PCC and LSTF tasks, respectively. A possible reason is that there are alternative contextualizing structures such as the convolutional neural network in the backbone models and the contribution of the attention mechanism is weakened. Besides, we also notice that the current efficient attention family has already surpassed vanilla attention in modeling noncausal self relationships on most tasks.

---

[4]The negative correlation here is biased due to limited exploration of causal and cross efficient attentions.

**Interpolation and Extrapolation on Long-Context Language Modeling**   Language models with efficient attentions are more likely to face unexpected long context that exceeds predefined lengths, resulting in an extrapolation problem. It is critical for efficient attentions to scale to longer sequences while preserving the performance on short ones. Therefore, we conduct an experiment on the language modeling task, where we train the model with the input of 8,192 tokens and calculate the perplexity of contexts with 256-16,384 tokens during the test. Figure 3c shows the results on both of interpolation and extrapolation. On one hand, as the size of context grows to 8,192, language models achieve decreasing perplexity. It means that longer contexts indeed improve language modeling, and efficient attentions do well in interpolation as expected. On the other hand, for sequences with more than 8,192 tokens, the perplexities of all these efficient attention-equipped language models become higher. Notably, local attention and LongShort with local features have only a slight drop in performance in extrapolation, while ABC with only global features fails to extrapolate to longer sequences. Although S4D performs well as causal attention, it has a large perplexity with sequence input longer than 8,192, indicating that S4D has relatively poor extrapolation ability.

## 5   RELATED WORK

**Long Sequence Modeling**   Most efficient attention architectures are designed for long sequence modeling. Previous works evaluate their proposed attention modules separately on Language Model (Choromanski et al., 2020; Qin et al., 2021; Peng et al., 2022), Masked Language Model (Qin et al., 2021; Peng et al., 2022), Image Classification (Choromanski et al., 2020; Zheng et al., 2022), and Machine Translation (Peng et al., 2022; 2021). Without unified criteria for evaluating attention modules, researchers find it hard to distinguish between them and select an appropriate attention module for their research. Later, Tay et al. (2020a) proposed Long Range Arena (LRA), a benchmark that contains 5 different classification tasks across language and image domainsHowever, their benchmark only considers the noncausal self attention pattern and contains most synthetic tasks which are far from real-world applications. Besides, different attention modules perform nearly equally on LRA (Xiong et al., 2022), which causes LRA a limited benchmark. Shaham et al. (2022) proposed SCROLLS for long language sequences but ignores long sequence modeling tasks in other fields.

**Efficient Attention**   The emergence and prevalence of Transformer models (Vaswani et al., 2017) as well as its strong capability for encoding and generation have imposed impressiveness on various applications. To improve the efficiency of attention in Transformer models, previous work explored efficient attention mechanisms, including sparse attention (Guo et al., 2019a; Ainslie et al., 2020; Beltagy et al., 2020), low-rank attention (Guo et al., 2019b; Chen et al., 2020; Xiong et al., 2021), kernel-based linearized attention (Choromanski et al., 2020; Katharopoulos et al., 2020), prototype attention (Vyas et al., 2020; Zhou et al., 2021), prior attention (Zhang et al., 2018; Ying et al., 2021; Tay et al., 2021), and memory compress (Liu* et al., 2018; Lee et al., 2019; Wang et al., 2020). For interested readers, we refer them to (Tay et al., 2020b) and (Lin et al., 2021) for a more detailed description of efficient attention.

## 6   CONCLUSION

In this paper, we propose to evaluate efficient attentions under a more fine-grained attention taxonomy with four distinguishable attention patterns, each of which reflects different attentive functionality. Under the taxonomy, we present Comprehensive Attention Benchmark (CAB), consisting of seven real-world tasks and eight backbone networks. We conduct exhaustive experiments to benchmark nine typical efficient attentions on CAB. The empirical results show that existing efficient attentions make significant progress in noncausal self attention, but are still struggling to catch up with vanilla attention in conditionality and causality modeling. The efficiency analysis shows that although existing efficient attentions enjoy lower computational complexity, most of them only start to achieve significant efficiency gain when applied to sequences with more than one thousand tokens, especially in the causal attention scenario. Moreover, we find that attention mechanisms, in the noncausal self case, are sometimes not helpful or even harmful to neural models that already contain feature mixing. Our study sheds light on long-context language modeling and shows that efficient attentions such as local attention and LongShort Transformer are promising for extrapolation problems in long-context language modeling.

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

## A   EFFICIENT ATTENTIONS WITH PERFORMANCE DIFFERENCES

**cosFormer**   requires a *cos*-reweighting mechanism to behave like vanilla softmax attention. However, its reweighting mechanism needs the maximum length between the source and target sequences in Eq. 16. Under causal self pattern, this operation requires different $L$ values for each token of $\mathbf{Q}$ and $\mathbf{K}$, which is ignored in its released codebase repository[5]. Therefore, the released code cannot model causality.

**Performer**   uses random feature matrices to project $\mathbf{Q}$ and $\mathbf{K}$. However, before transforming $\mathbf{Q}, \mathbf{K}$ with the softmax kernel, its released codebase[6] uses the golden target sequence length to compute random feature maps under causal self pattern. However, this operation brings inconsistency between training and inference, because the golden target length is unavailable during inference. Thus, we cannot use the released code to perform causal self tasks.

## B   INCREASING COMPUTATION COSTS FOR CAUSAL EFFICIENT ATTENTION

In this section, we use an efficient attention example RFA (Peng et al., 2021) to illustrate enforcing causality might bring an increase in computation for efficient attention. In RFA, the noncausal version models the query $\boldsymbol{q}_t$ as:

$$\text{RFA}(\boldsymbol{q}_t, \{\boldsymbol{k}_i\}, \{\boldsymbol{v}_i\}) = \frac{\phi(\boldsymbol{q}_t)^\top \sum_i \phi(\boldsymbol{k}_i) \otimes \boldsymbol{v}_i}{\phi(\boldsymbol{q}_t) \sum_j \phi(\boldsymbol{k}_j)} \tag{7}$$

$$= \frac{\phi(\boldsymbol{q}_t)^\top \mathbf{S}_t}{\phi(\boldsymbol{q}_t) \cdot \mathbf{z}_t} \tag{8}$$

where $\mathbf{S}_t \in \mathbb{R}^{2D \times d}$ and $\mathbf{z}_t \in \mathbb{R}^d$ and $2D$ is the dimensionality of kernel $\phi$. In its causal version, the hidden state $\mathbf{S}_t$ and $\mathbf{z}_t$ are recurrently modified to maintain the causality:

$$\mathbf{S}_t = \mathbf{S}_{t-1} + \phi(\boldsymbol{k}_t) \otimes \boldsymbol{v}_t \quad \mathbf{z}_t = \mathbf{z}_{t-1} + \phi(\boldsymbol{k}_t) \tag{9}$$

We need to store $\mathbf{S}_t$ and $\mathbf{z}_t$ explicitly during training, which contains the historical information specific for each query $\boldsymbol{q}_t$. As a result, these recurrent operations add additional $O(ndD)$ time and memory consumption.

## C   COMPOSITIONAL INDEX

We define $s_{ij}^k$ as the score of $i$-th attention model on $k$-th metric in the $j$-th task, where $k = 1, \ldots, K$ and $j = 1, \ldots, M$. To obtain Compositional index (*CI*), we first normalize $s_{ij}^k$. For the metric whose value is higher indicating high performance, $s_{ij}^k$ is normalized to $\overline{s}_{ij}^k$ by $\overline{s}_{ij}^k = \frac{s_{ij}^k - \mu_j^k}{\sigma_j^k}$, where $\mu_j^k$ and $\sigma_j^k$ denote the mean and standard deviation of the models' scores on this metric. Otherwise, we normalize $s_{ij}^k$ by $\overline{s}_{ij}^k = \frac{\mu_j^k - s_{ij}^k}{\sigma_j^k}$ for the metric whose value is lower indicating high performance. Then we average the normalized score of each metric to obtain a task score $CI_{ij} = \frac{\sum_{k=1}^K \overline{s}_{ij}^k}{K}$, which is further converted to the final score $CI_i = \frac{\sum_{j=1}^M CI_{ij}}{M}$.

## D   HYPERPARAMETERS

Hyperparameters for all tasks are shown in Table 4. We also report the hyperparameters of efficient attentions in Table 5.

---

[5] https://github.com/OpenNLPLab/cosFormer/blob/main/cosformer.py#L106
[6] https://github.com/google-research/google-research/blob/master/performer/fast_attention/tensorflow/fast_attention.py#L190

Table 4: Hyperparameters for the tasks in `CAB`.

| Task | TTS | LSTF | | | PCC | Sumn | LM | MLM | SR |
|---|---|---|---|---|---|---|---|---|---|
| Data | LJSpeech | ETT | ECL | Weather | PCN | Multi-News | PG-19 | | FFHQ CelebA-HQ |
| Batch Size | 48 | 32 | | | 48 | 64 | 16 | 256 | 4 |
| Number of Steps | 20K | 6 (epochs) | | | 300 (epochs) | 50K | 125K | 125K | 1M |
| Warmup Steps | 4K | - | | | 4K | 1K | 10K | 5K | - |
| Peak Learning Rate | 5e-4 | 1e-4 | | | 5e-4 | 5e-4 | 5e-4 | 3e-4 | 1e-4 |
| Scheduler | Inverse Sqrt | Exponential Decay | | | LamdaLR | Inverse Sqrt | Inverse Sqrt | Polynomial Decay | Linear |
| Optimizer | AdamW | AdamW | | | AdamW | AdamW | AdamW | AdamW | AdamW |
| Adam | (0.9, 0.98) | (0.9, 0.999) | | | (0.9, 0.98) | (0.9, 0.98) | (0.9, 0.999) | (0.9, 0.98) | (0.9, 0.999) |
| Clip Norm | 5.0 | 0.0 | | | 0.0 | 0.0 | 0.0 | 0.0 | 0.0 |
| Attention Dropout | 0.1 | 0.05 | | | 0.0 | 0.1 | 0.1 | 0.1 | 0.2 |
| Weight Decay | 0.01 | 0.0 | | | 5e-4 | 0.01 | 0.01 | 0.01 | 0.0 |
| Tokens per Batch | - | - | | | - | - | $2^{17}$ | $2^{19}$ | - |
| Iteration | - | 5 | 3 | 3 | - | - | - | - | - |

Table 5: Hyperparameters for various efficient attentions under noncausal self attention pattern (a), causal self attention pattern (b), noncausal cross attention pattern (c), and causal cross attention pattern (d). (*) We find that the performance of LongShort with num_landmarks = 64 is significantly lower than its performance with num_landmarks = 16 on Sum task under the noncausal self attention pattern. So we finally use LongShort with num_landmarks = 16.

| Method | Hyperparameters | TTS | Sum | SR | MLM |
|---|---|---|---|---|---|
| local | wsize | 15 | 15 | 15 | 15 |
| ABC | num_landmarks | 64 | 64 | 16 | 16 |
| LARA | num_landmarks | 64 | 64 | 16 | 16 |
| ProbSparse | factor | 5 | 5 | 5 | 5 |
| Performer | approx_attn_dim | 64 | 64 | 16 | 16 |
| LongShort | wsize | 15 | 15 | 15 | 15 |
| LongShort | num_landmarks | 64 | 16* | 16 | 16 |
| LongShort | conv_kernel_size | 5 | - | - | - |
| Nyströmformer | num_landmarks | 64 | 64 | 16 | 16 |
| Nyströmformer | conv_kernel_size | 5 | - | - | - |
| S4D | d_state | 64 | 64 | 16 | 16 |
| cosFormer | - | | - | | |

(a)

| Method | Hyperparameters | TTS Sum LM |
|---|---|---|
| local | wsize | 16 |
| LongShort | num_landmarks | 64 |
| LongShort | wsize | 16 |
| ABC | num_landmarks | 64 |

(b)

| Method | Hyperparameters | PCC | LSTF |
|---|---|---|---|
| ABC | num_landmarks | 64 | 16 |
| Performer | approx_attn_dim | 64 | 16 |
| cosFormer | - | - | - |

(c)

| Method | Hyperparameters | TTS Sum |
|---|---|---|
| ABC | num_landmarks | 64 |
| Performer | approx_attn_dim | 64 |

(d)

# E  TASK SCORE FOR COMPARISON

We show the compositional index of each task in Table 6. For a new efficient attention, it can use our provided means and variances to obtain the normalized score. The means and standard deviations of metrics are shown in the Table 7, Table 8, Table 9, and Table 10.

# F  DETAILED DESCRIPTION OF EFFICIENT ARCHITECTURES

We here only present the noncausal self forms of these efficient architectures. In the subsequent discussion, we denote $n$ and $m$ as the target and source sequence lengths, respectively. Also, we omit the factor $1/\sqrt{d}$ for simplicity.

**Local Attention**   Local attention (Luong et al., 2015) intuitively forces each query token to attend to its neighborhood tokens within a fixed window size $w$ to reduce the computation costs. In the full attention map, the neighborhood tokens contribute a large proportion of the attention score for the querying token (Qin et al., 2021). Therefore, local attention intuitively selects $w$ ($w/2$ for causal

Table 6: Compositional index.

| Method | NS | | | | CS | | | NC | | CC | |
|--------|-----|-----|-----|-----|-----|-----|-----|-----|------|-----|-----|
| | TTS | Sum | SR | MLM | TTS | Sum | LM | PCC | LSTF | TTS | Sum |
| vanilla | -0.301 | -0.246 | -0.077 | 0.527 | -0.047 | 0.784 | - | 0.449 | 0.744 | 1.047 | 0.867 |
| local | 0.362 | 2.617 | 0.421 | 0.515 | -1.361 | 0.337 | 0.305 | | | | |
| ABC | 0.043 | -0.678 | -2.525 | 0.414 | 0.707 | -1.461 | -1.117 | 0.573 | -0.164 | -0.112 | 0.228 |
| LARA | -0.639 | -0.522 | 0.554 | 0.479 | | | | | | | |
| cosFormer | 0.516 | -0.190 | 1.042 | 0.498 | | | | -1.496 | -0.135 | | |
| ProbSparse | 0.705 | -0.246 | -0.697 | -2.408 | | | | | | | |
| Performer | -0.282 | -0.088 | 0.439 | -1.211 | | | | 0.473 | -0.445 | -0.935 | -1.094 |
| LongShort | 0.572 | -0.322 | 0.298 | 0.528 | 0.701 | 0.340 | 0.812 | | | | |
| Nyströmformer | -2.011 | -0.321 | 0.088 | 0.481 | | | | | | | |
| S4D | 1.035 | - | 0.457 | 0.176 | 1.030 | 1.143 | 0.780 | | | | |

self attention pattern) tokens around a querying token to serve as $\mathbf{K}_w$ and $\mathbf{V}_w$. Its formulation is as follows:

$$\text{Local}(\boldsymbol{q}_i, \mathbf{K}_w, \mathbf{V}_w) = \text{softmax}(\boldsymbol{q}_i \mathbf{K}_w^\top) \mathbf{V}_w \tag{10}$$

where $\mathbf{K}_w, \mathbf{V}_w \in \mathbb{R}^{w \times d}$ denotes the neighborhood tokens within $w$ size around the query token $\boldsymbol{q}_i$ in $\mathbf{K}$ and $\mathbf{V}$. The complexity is of linear complexity $O(nwd)$ when $w$ is predefined and independent of sequence length $n$. Due to the locality inductive bias, local attention is competent only on causal self and noncausal self attention patterns.

**Nyströmformer** Nyströmformer (Xiong et al., 2021) adopts the Nyström method to approximate the softmax attention matrix. It selects $r$ landmarks for $\mathbf{Q}$ and $\mathbf{K}$ with Segment-means (Shen et al., 2018) and produces $\widetilde{\mathbf{Q}}$ and $\widetilde{\mathbf{K}}$. Using iterative Moore-Penrose pseudoinverse (Razavi et al., 2014), Nyströformer decomposes the softmax attention matrix into three sub-softmax matrices as follows:

$$\text{Nyströmformer}(\mathbf{Q}, \mathbf{K}, \mathbf{V}) = \text{softmax}(\mathbf{Q}\widetilde{\mathbf{K}}^\top)(\text{softmax}(\widetilde{\mathbf{Q}}\widetilde{\mathbf{K}}^\top))^+ \text{softmax}(\widetilde{\mathbf{Q}}\mathbf{K}^\top)\mathbf{V} \tag{11}$$

where $(\text{softmax}(\widetilde{\mathbf{Q}}\widetilde{\mathbf{K}}^\top))^+$ is the Moore-Penrose pseudoinverse of $\text{softmax}(\widetilde{\mathbf{Q}}\widetilde{\mathbf{K}}^\top)$. The complexity Moore-Penrose pseudoinverse is linear with respect to $n$ when its iteration time is constant. Therefore, the overall computation cost is $O(nrd)$ when $r$ is predefined. Because it needs to select the same number of landmarks for $\mathbf{Q}$ and $\mathbf{K}$ and requires the full $\mathbf{Q}$ representation, it is only adaptable in the noncausal self attention pattern.

**Performer** Performer (Choromanski et al., 2020) is another linear transformer using *random feature map* $\phi(\cdot) : \mathbb{R}^d \to \mathbb{R}^r_+$ (for some $r > 0$) as a kernel function to reduce computational complexity. Formally, the random feature map is defined to approximate the exponential kernel as

$$\exp(\boldsymbol{q}_i^\top \boldsymbol{k}_j) = \mathbb{E}_q[\phi(\boldsymbol{q}_i)^\top \phi(\boldsymbol{k}_j)], \tag{12}$$

where $\phi(\boldsymbol{x}) = \exp(W_r \boldsymbol{x} - \frac{1}{2}\|\boldsymbol{x}\|^2 \mathbf{1}_r) \in \mathbb{R}^r$ and all elements of $W_r$ are independently drawn from a standard Gaussian $q(w) = \mathcal{N}(w; 0, 1)$. Given the defined random feature map, the whole softmax attention for each query can then be approximated as

$$\sum_{j=1}^m \frac{\exp(\boldsymbol{q}_i^\top \boldsymbol{k}_j)\boldsymbol{v}_j}{\sum_{j'=1}^m \exp(\boldsymbol{q}_i^\top \boldsymbol{k}'_j)} \approx \frac{\phi(\boldsymbol{q}_i)^\top \sum_{j=1}^m \phi(\boldsymbol{k}_j)\boldsymbol{v}_j^\top}{\phi(\boldsymbol{q}_i)^\top \sum_{j'=1}^m \phi(\boldsymbol{k}'_j)} = \text{Performer}(\boldsymbol{q}_i, \mathbf{K}, \mathbf{V}) \tag{13}$$

Thanks to the kernel approximation, Performer achieves $O(nrd)$ complexity in both noncausal self and noncausal cross attention patterns; however, for causal self and causal cross attentions, Performer has to store the cumulative summation for both $\sum_{j=1}^m \phi(\boldsymbol{k}_j)\boldsymbol{v}_j^\top$ and $\sum_{j=1}^m \phi(\boldsymbol{k}_j)$, which requires a dedicated CUDA operator to accelerate the computation.

**LARA** LARA (Zheng et al., 2022) further extends the random feature approximation from estimating individual exponential kernels to directly estimating the whole softmax function. In particular, there exists a distribution $p$ that instantiates the random feature (Zheng et al., 2022) such as:

$$\sum_{j=1}^m \frac{\exp(\boldsymbol{q}_i^\top \boldsymbol{k}_j)\boldsymbol{v}_j^\top}{\sum_{j'=1}^m \exp(\boldsymbol{q}_i^\top \boldsymbol{k}'_j)} = \mathbb{E}_p\left[\sum_{j=1}^m \frac{\phi(\boldsymbol{q}_i)^\top \phi(\boldsymbol{k}_j)\boldsymbol{v}_j^\top}{\sum_{j'=1}^m \phi(\boldsymbol{q}_i)^\top \phi(\boldsymbol{k}'_j)}\right]. \tag{14}$$

Table 7: The means and standard deviations for noncausal self attention pattern evaluation.

| Method | TTS | | | | Sum | | | SR | | MLM |
| | FastSpeech 2 | | Transformer-TTS | | Transformer | | | SR3 | | RoBERTa |
| | MCD | MSD | MCD | MSD | R-1 | R-2 | R-L | PSNR | SSMI | PPL |
| --- | --- | --- | --- | --- | --- | --- | --- | --- | --- | --- |
| $\mu$ | 3.425 | 1.974 | 4.069 | 2.190 | 34.89 | 6.79 | 31.96 | 23.21 | 0.676 | 36.292 |
| $\sigma$ | 0.068 | 0.036 | 0.094 | 0.038 | 1.396 | 1.410 | 1.314 | 0.274 | 0.015 | 62.308 |

Table 8: The means and standard deviations for causal self attention pattern evaluation.

| Method | TTS | | Sum | | | LM |
| | Transformer-TTS | | Transformer | | | GPT-2 |
| | MCD | MSD | R-1 | R-2 | R-L | PPL |
| --- | --- | --- | --- | --- | --- | --- |
| $\mu$ | 4.083 | 2.200 | 32.96 | 6.09 | 30.26 | 22.26 |
| $\sigma$ | 0.045 | 0.013 | 1.927 | 0.410 | 1.619 | 8.299 |

By viewing softmax attention as an expectation over a potentially complicated distribution $p$, Zheng et al. (2022) noted that Performer can be understood as performing self-normalized importance sampling to approximate this expectation, with standard Gaussian as the proposal distribution. LARA generalizes this view by adopting multiple adaptive proposal distributions to further improve the approximation fidelity. It is shown that LARA takes the following form:

$$\text{LARA}\,(\boldsymbol{q}_n, \mathbf{K}, \mathbf{V}) = \frac{\phi(\boldsymbol{q}_n)^\top A_n \sum_{j=1}^m \phi(\boldsymbol{k}_j)\boldsymbol{v}_j^\top}{\phi(\boldsymbol{q}_n)^\top A_n \sum_{j'=1}^m \phi(\boldsymbol{k}_{j'})}, \tag{15}$$

where $A_n$ is a query-specific diagonal matrix that ensures the computation overall still yields a valid self-normalized importance sampling estimation; and the elements of sampled $W_r$ in $\phi(\boldsymbol{x}) = \exp(W_r\boldsymbol{x} - \frac{1}{2}\|\boldsymbol{x}\|^2\mathbf{1}_r)$ are allowed to be drawn from different distributions beyond standard Gaussian.

**cosFormer** Following the same method defined in Eq. 13, cosFormer (Qin et al., 2021) adopts another kernel function $\text{ReLU}(\cdot)$ instead of random feature map. Due to the importance of the softmax operator that can ensure the non-negativeness of value weights and stabilize the training process, cosFormer designed a new *cos*-based reweighting mechanism similar to softmax. Denote $\boldsymbol{q}_i' = \text{ReLU}(\boldsymbol{q}_i)$ and $\boldsymbol{k}_j' = \text{ReLU}(\boldsymbol{k}_j)$, the re-weight formula is expressed as follows:

$$s(\boldsymbol{q}_i, \boldsymbol{k}_j) = \boldsymbol{q}_i^\top \boldsymbol{k}_j \cos(\frac{\pi}{2} \times \frac{i-j}{L}) \tag{16}$$

where $L = \max(n, m)$. Using this formula and Ptolemy's theorem, the final expression of cosFormer is as follows:

$$\text{cosFormer}(\boldsymbol{q}_i, \mathbf{K}, \mathbf{V}) = \frac{\sum_{j=1}^m \boldsymbol{q}_i^{\cos}((\boldsymbol{k}_j^{\cos})^\top \boldsymbol{v}_j) + \sum_{j=1}^m \boldsymbol{q}_i^{\sin}((\boldsymbol{k}_j^{\sin})^\top \boldsymbol{v}_j)}{\sum_{j=1}^m (\boldsymbol{q}_i^{\cos})^\top \boldsymbol{k}_j^{\cos} + \sum_{j=1}^m (\boldsymbol{q}_i^{\sin})^\top \boldsymbol{k}_j^{\sin}} \tag{17}$$

where $\boldsymbol{q}_i^{\cos} = \boldsymbol{q}_i \cos(\frac{\pi i}{2L}), \boldsymbol{q}_i^{\sin} = \boldsymbol{q}_i \sin(\frac{\pi i}{2L}), \boldsymbol{k}_j^{\cos} = \boldsymbol{k}_j \cos(\frac{\pi j}{2L}), \boldsymbol{k}_j^{\sin} = \boldsymbol{k}_j \sin(\frac{\pi j}{2L})$. Similar to Performer, it can efficiently fulfill noncausal self and noncausal cross attentions with linear complexity $O(nd^2)$, but its causal attention relies on a specialized CUDA operator to achieve high efficiency.

**LongShort Transformer** LongShort Transformer (Zhu et al., 2021) models the long-range and short-term dependency of a sequence with two different methods discussed above. For long-range modeling, it utilizes the low-rank property of the softmax function to convert key and value to a more compact representation using a learnable matrix $W^P \in \mathbb{R}^{d \times r}$: $\overline{\mathbf{K}} = P^\top \mathbf{K} \in \mathbb{R}^{r \times d}$ and $\overline{\mathbf{V}} = P^\top \mathbf{V} \in \mathbb{R}^{r \times d}$ where $P = \text{softmax}(\mathbf{K}W^P)$ and $r$ is the compressed dimension. For short-term modeling, it borrows local attention to output a token representation fused with neighborhood features via segment-wise local attention. Let $l$ be the segment length and $s_i = \lfloor i/l \rfloor$ be the segment id for $i$-th token, segment-wise local attention produces $\mathbf{K}_w = \mathbf{K}_{(s_i-1)l+\frac{l}{2}:(s_i+1)l+\frac{l}{2}}$

Table 9: The means and standard deviations for noncausal cross attention pattern evaluation.

| Method | PCC | | | LSTF | | | | | |
| | PoinTr | | | Informer | | | | | |
| | | | | ETT | | Weather | | ECL | |
| | CD-$\ell_1$ | CD-$\ell_2$ | F-score | MSE | MAE | MSE | MAE | MSE | MAE |
| --- | --- | --- | --- | --- | --- | --- | --- | --- | --- |
| $\mu$ | 7.702 | 0.247 | 0.762 | 1.189 | 0.806 | 0.476 | 0.511 | 0.620 | 0.581 |
| $\sigma$ | 0.647 | 0.033 | 0.039 | 0.056 | 0.021 | 0.010 | 0.006 | 0.189 | 0.103 |

Table 10: The means and standard deviations for causal cross attention pattern evaluation.

| Method | TTS | | Sum | | |
| | Transformer-TTS | | Transformer | | |
| | MCD | MSD | R-1 | R-2 | R-L |
| --- | --- | --- | --- | --- | --- |
| $\mu$ | 5.503 | 2.627 | 31.35 | 5.26 | 28.80 |
| $\sigma$ | 1.292 | 0.427 | 3.771 | 1.260 | 3.286 |

and $\mathbf{V}_w = \mathbf{V}_{(s_i-1)l+\frac{l}{2}:(s_i+1)l+\frac{l}{2}}$. Combined with these two methods, the output of LongShort Transformer is formalized as follows:

$$\text{LongShort}(\boldsymbol{q}_i, \mathbf{K}, \mathbf{V}) = \text{softmax}(\boldsymbol{q}_i[\mathbf{K}_w; \overline{\mathbf{K}}]^\top)[\mathbf{V}_w; \overline{\mathbf{V}}] \qquad (18)$$

where $[\cdot; \cdot]$ is a concatenation operator at the first dimension. The complexity $O(nrd + nwd)$ is also linear *w.r.t.* the target length $n$. Due to its introduction of local attention, LongShort Transformer can only complete tasks under causal self and noncausal self attention patterns.

**ProbSparse Attention** ProbSparse (Zhou et al., 2021) calculates attention scores for few $(\mathbf{Q}, \mathbf{K})$ dot-product pairs based on sparsity of self attention scores. Specifically, ProbSparse selects some "important" queries whose attention distributions on keys are non-uniform. Then we only need to calculate the attention scores between the "important" queries and all keys and output the weighted sum of $\mathbf{V}$. For other trivial queries, ProbSparse takes the mean of $\mathbf{V}$ as outputs.

To distinguish the "important" queries, ProbSparse first samples some keys $\overline{\mathbf{K}}$, and then measures the sparsity of query $q_i$ by $\bar{M}(\mathbf{q}_i, \overline{\mathbf{K}})$ as follows:

$$\bar{M}(\mathbf{q}_i, \overline{\mathbf{K}}) = \max_j \left\{ \frac{\mathbf{q}_i \bar{\mathbf{k}}_j^\top}{\sqrt{d}} \right\} - \frac{1}{L_K} \sum_{j=1}^{L_K} \frac{\mathbf{q}_i \bar{\mathbf{k}}_j^\top}{\sqrt{d}} \qquad (19)$$

where $L_K$ denotes the number of keys. Finally, the Top-$u$ queries with large $\bar{M}(\mathbf{q}_i, \overline{\mathbf{K}})$ are the "important" queries.

**ABC** In this paper, we use ABC to denote $\text{ABC}_{\text{MLP}}$ (Peng et al., 2022). $\text{ABC}_{\text{MLP}}$ bounds the amount of memory slot for $\mathbf{K}$ and $\mathbf{V}$ to be $r < m$, independent of both $n$ and $m$. $\text{ABC}_{\text{MLP}}$ uses a learnable matrix to fill each memory slot with reweighted key and value representations:

$$\widetilde{\mathbf{K}} = \text{softmax}(W_\phi^K \mathbf{K}^\top)\mathbf{K}, \quad \widetilde{\mathbf{V}} = \text{softmax}(W_\phi^V \mathbf{V}^\top)\mathbf{V}$$

where $W_\phi^K, W_\phi^V \in \mathbb{R}^{r \times d}$ is a learnable matrix representing the bounded memory. Leveraging the bounded representation $\widetilde{\mathbf{K}}$ and $\widetilde{\mathbf{V}}$, $\text{ABC}_{\text{MLP}}$ executes following softmax attention:

$$\text{ABC}_{\text{MLP}}(\boldsymbol{q}_i, \mathbf{K}, \mathbf{V}) = \text{softmax}(\boldsymbol{q}_i \widetilde{\mathbf{K}}^\top)\widetilde{\mathbf{V}} \qquad (20)$$

$\text{ABC}_{\text{MLP}}$ is again of linear complexity $O(nrd)$.

**S4D** S4D (Gu et al., 2022) is different from the above attention-based methods, as it relies on State Space Matrix called "HiPPO matrix" (Gu et al., 2020) which enables the whole model to be viewed as

a convolutional model. It does not require the introduction of $\mathbf{Q}, \mathbf{K}, \mathbf{V}$ as it takes the whole modeling process as a linear time-invariant (LTI) system $u(t) \mapsto y(t)$ where $u(t)$ is the input sequence. To parameterize the system, S4D designs a Convolution Kernel $K(t) \in \mathbb{R}^{n \times d}$ and introduces a inner state dimension $r$ during parameterization of $K(t)$:

$$K(t) = \boldsymbol{C}e^{t\boldsymbol{A}}\boldsymbol{B}, \quad y(t) = (K * u)(t)$$

where $\boldsymbol{A}, \boldsymbol{B}, \boldsymbol{C}$ are all initialized with certain values and trained with the whole model. During implementation, S4D processes $u$ and $K$ in the frequency domain and transforms back the output to the time domain:

$$K(\omega) = \texttt{fft}(K(t)) \tag{21}$$
$$U(\omega) = \texttt{fft}(u(t)) \tag{22}$$
$$S4D(u) = \texttt{ifft}(K(\omega)U(\omega)) \tag{23}$$

Therefore, S4D has the complexity of $O(nrd)$.

## G  EFFICIENCY LENGTH

We show efficiency lengths under noncausal self attention pattern in Table 11, where we set hyper-parameters as $\texttt{emb\_dim} = 512$, $\texttt{head} = 8$, $\texttt{batch\_size} = 4$, $\texttt{wsize} = 16$, $\texttt{num\_landmarks} = 16$, $\texttt{approx\_attn\_dim} = 16$, $\texttt{d\_state} = 16$, $\texttt{conv\_kernel\_size} = 5$. We use the adaptive factor for ProbSparse introduced in Skyformer (Chen et al., 2021b) repository[7].

Table 11: The efficiency lengths of attention architectures compared to vanilla attention under noncausal self attention pattern.

| Method | Running Time | | Memory Usage | |
|---|---|---|---|---|
| | Efficiency Length | $R^2$ | Efficiency Length | $R^2$ |
| Performer | 2,361 | 0.977 | 37 | 0.999 |
| LARA | 2,541 | 0.975 | 68 | 0.999 |
| ABC | 1,877 | 0.964 | 70 | 0.999 |
| Nyströmformer | 3,045 | 0.987 | 94 | 0.999 |
| cosFormer | 2,539 | 0.990 | 164 | 0.999 |
| ProbSparse | 3,450 | 0.989 | 281 | 0.994 |
| LongShort | 5,652 | 0.998 | 342 | 0.999 |
| S4D | 6,011 | 0.996 | 949 | 0.999 |
| local | 4,195 | 0.996 | 1,169 | 0.999 |

## H  BENEFIT OF ATTENTION

We report the CI of backbone models with and without attention in Table 12.

Table 12: The CI of backbone models with and without attention on tasks. '-' denotes that the method fails on the task.

| Method | TTS | Sum | PCC | SR | LSTF | MLM |
|---|---|---|---|---|---|---|
| efficient attention | 1.034 | 0.715 | 0.720 | 1.041 | 0.591 | 0.528 |
| vanilla attention | -0.300 | -0.246 | 0.908 | -0.076 | -0.102 | 0.527 |
| w/o attention | -2.155 | -1.817 | 0.940 | -0.058 | 0.027 | – |

---

[7]https://github.com/pkuzengqi/Skyformer

# I INTER-TASK CORRELATION

Figure 4 shows the task-to-task correlation. Except for the causal cross attention pattern, tasks within the other three patterns scarcely correlate positively with each other. This suggests that there does hardly exist a single task that can comprehensively test the capability of some attention module under an attention pattern.

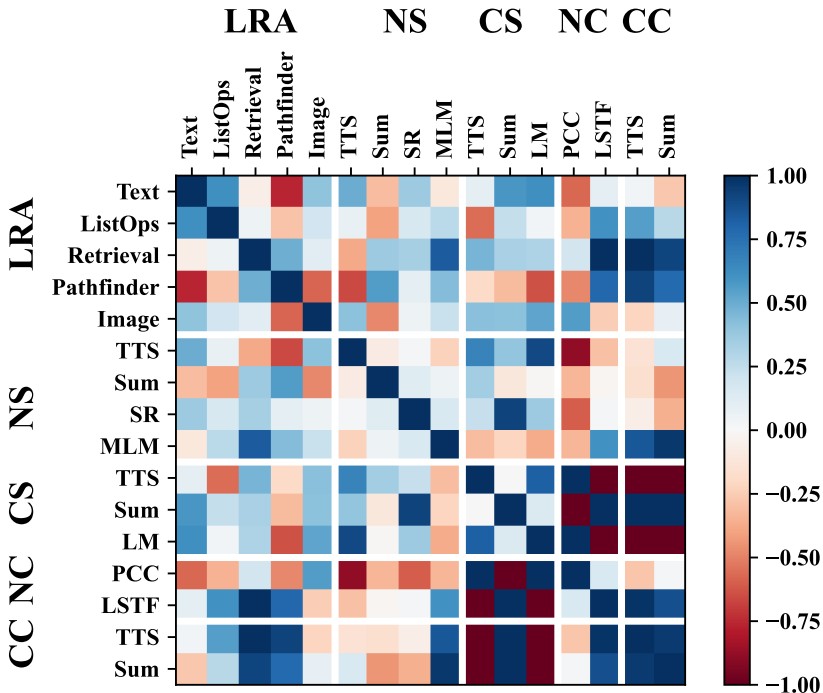

Figure 4: Task-to-task correlation.

