# OpenReview forum: "CAB: Comprehensive Attention Benchmarking on Long Sequence Modeling"
_ICLR.cc/2023/Conference — Submitted to ICLR 2023_

### Official Review · Reviewer_VyuY · 2022-10-22

**Confidence:** 4
**Correctness:** 3
**Technical Novelty And Significance:** 2
**Empirical Novelty And Significance:** 2
**Recommendation:** 6

**Clarity, Quality, Novelty And Reproducibility:**

Clarity: The paper is clear to follow, but could use some copy editing. I think the focus on distinguishing self-attention from causal/cross/etc attention is a little bit misplaced.

Quality: I think the most useful contribution of this paper is simply aggregating more long-sequence benchmarks as "standard." The categorization of methods into self-attention, causal, cross, etc seems to have some basic mistakes.

For example, S4D is inherently a causal model (this is the definition of a state space model - it models how a state changes through time). The "bidirectional" or noncausal version of S4D is a modification on top. I am not as familiar with the details of the other methods, but this category error makes me concerned.

For a benchmark paper, it would be nice to see more baselines for exact attention, such as FlashAttention.

One question: which tokenizer do you use for PG-19? Making that clear up front would be helpful (it is one of the reasons that existing work is a little hard to compare).

Novelty: I think this type of benchmark has been missing from the literature, so it is welcome. However, the analysis and evaluation on top are a little bit lacking.

Reproducibility: Seems clear.

**Strength And Weaknesses:**

+ Fills in a missing gap in the literature beyond LRA
+ Will be useful for researchers evaluating long-sequence models

- There appear to be a few basic mistakes in the methods characterization
- It would be nice to have a few more baselines
- Paper could use some copy edits

**Summary Of The Paper:**

This paper presents CAB, a new benchmark to test Transformers and other sequence models on long sequence data. The paper benchmarks a number of methods as a baseline.

**Summary Of The Review:**

I think this paper could make a good contribution to the literature, but the paper has a few basic mistakes that should be corrected. If my concerns are addressed in the rebuttal, I'm happy to raise my score above the acceptance threshold.

Update after rebuttal: my concerns have been addressed, so I am increasing my score to a 6.

---

> ### Author Response · Authors · 2022-11-08
> **Response to Reviewer VyuY**
>
> Thanks for your helpful comments! We update the revised paper and respond to your questions below:
>
> > Q1: S4D is inherently a causal model (this is the definition of a state space model - it models how a state changes through time). The "bidirectional" or noncausal version of S4D is a modification on top. I am not as familiar with the details of the other methods, but this category error makes me concerned.
>
> A1: Thank you for pointing out that. We modified Table 2 and added the results of S4D under the causal self attention pattern.  We show the modified Table 3b as follows:
>   |  Method | TTS MCD↓ | TTS MSD↓ | Sum R-1↑ | Sum R-2↑ | Sum R-L↑ | LM PPL↓ | CI ↑
>   | ---- | ---- | ---- |  ---- | ---- |  ---- | ---- |  ---- |
>   |  vanilla        | 4.095 | 2.198 | 34.61 | 6.35 | 31.66 |     -    |     -      |
>   |  S4D           | **4.030** | **2.188** | **34.90** | **6.65** | **31.98** | 15.78 | **0.985**  |
>   |  LongShort  | 4.039 | 2.195 | 33.55 | 6.27 | 30.71 | **15.52** | 0.617  |
>   |  local           | 4.141 | 2.220 | 33.50 | 6.27 | 30.74 | 19.73 | -0.239 |
>   |  ABC           | 4.058 | 2.189 | 30.17 | 5.48 | 27.92 | 29.13 | -0.623 |
>
> Experimental results show that the state space model is more suitable for causal self attention pattern. Moreover, we also evaluate the interpolation and extrapolation of S4D on long-context language modeling. We show the perplexity of efficient attentions based on different context length in the LM task as follows:
>   |  Method  | 256 | 512 | 1024 |  2048 |  4096 |  8192 | 12288  | 16384 |
>   | ---- | ---- | ---- | ---- | ---- | ---- | ---- | ---- | ---- |
>   |  S4D           | 20.71 | 18.81 | 17.57 | 16.73 | 16.17 | 15.78 | 51.96 |  fail   |
>   |  LongShort  | 19.85 | 18.16 | 17.06 | 16.33 | 15.85 | 15.52 | 19.17 | 26.06 |
>   |  local           | 21.25 | 20.44 | 20.05 | 19.86 | 19.77 | 19.73 | 21.24 | 22.16 |
>   |  ABC           | 36.87 | 34.61 | 33.22 | 32.31 | 31.67 | 31.53 | 86.78 | 147.43 |
>
> We can find that although S4D performs well as causal attention, it has a large perplexity with more than 8,192 tokens, indicating that S4D has relatively poor extrapolation ability.
>
> > Q2: Which tokenizer do you use for PG-19?
>
> A2: We use the BPE-based GPT-2 tokenizer for PG-19, which is consistent with the model pre-training approach.
>
> > Q3: For a benchmark paper, it would be nice to see more baselines for exact attention, such as FlashAttention.
>
> A3: We consider covering as many attention mechanisms with different philosophies as possible when choosing the baseline attention mechanism. We will progressively include new efficient attentions (e.g., FlashAttention) in the future.

---

> > ### Comment · Reviewer_VyuY · 2022-11-16
> > **Response**
> >
> > Thank you for these updates, they have addressed my concerns, and I will be increasing my score.
> >
> > Re FlashAttention - it is an exact attention implementation, so the results on the benchmark tasks should be the same as vanilla attention. The only difference will be in the memory footprint and runtime (memory grows linearly with sequence length).

---

> > > ### Author Response · Authors · 2022-11-18
> > > **Response to Reviewer VyuY**
> > >
> > > Thank you very much for your insightful feedback on the paper. We're glad our responses addressed your concerns.

---

### Official Review · Reviewer_MeQL · 2022-10-23

**Confidence:** 4
**Correctness:** 3
**Technical Novelty And Significance:** 2
**Empirical Novelty And Significance:** 2
**Recommendation:** 5

**Clarity, Quality, Novelty And Reproducibility:**

Clarity: the paper is well written and it is clear.

Quality: The quality of this work is good.

Novelty: The novelty is limited.

Reproducibility: unclear at this time.

**Strength And Weaknesses:**

Strength
1. the paper is well written and feeling pleasure to read. It summarize many eixting of Transformer.

2. The authors propose Comprehensive Attention Benchmark for ling sequence modeling.

Weaknessness
1. the novelty of this work is not enough for ICLR. It seems that this paper is more related to benchmark dataset.

2. They claim that  related codes will be released at https://github.com/Anonymous. However, I cannot find it.


**Summary Of The Paper:**

In this paper, the authors propose Comprehensive Attention Benchmark (CAB) under a fine-grained attention taxonomy with four distinguishable attention patterns, namely, noncausal self, causal self, noncausal cross, and causal cross attentions.


**Summary Of The Review:**

This paper proposes to evaluate efficient attentions under a more fine-grained attention taxonomy
with four distinguishable attention patterns, each of which reflects different attentive functionality.

---

> ### Author Response · Authors · 2022-11-08
> **Response to Reviewer MeQL**
>
> Thank you for taking the time to review our paper. We respond to your questions below:
>
> > Q1: the novelty of this work is not enough for ICLR. It seems that this paper is more related to benchmark dataset.
>
> A1: Our contribution mainly consists of two parts: 1. A new benchmark based on a fine-grained attention taxonomy to evaluate efficient attentions; 2. Investigation on important problems of efficient attention to provide insight for the design of higher performance attention mechanisms in the future. Specifically:
>   1. CAB shed light on finegrained attention taxonomy, where efficient cross attentions and causal attentions are largely under-explored. The previous work LRA partially focuses on noncausal self attention, resulting in the development of efficient cross attention and causal attention lagging behand noncausal self attentions. We argue that efficiency of cross attention and causal attention are equally important to noncausal self one, and should be fairly studied. Efficient cross attention can help aggregation of non-homologous information and improve conditionality modeling tasks, such as data-to-text generation [1,2,3] and knowledge graph-based generation [4,5]. And efficient causal attention brings up strong generation capability and benefits generation tasks, such as text generation [6,7], language modeling[8,9], and text-to-speech synthesis [10].
>   2. Through the lens of CAB, we investigate fundamental problems of efficient attention that haven't been studied before. Efficiency length quantifies practical cost of efficient attention beyond theoretical complexity, which is critical to real-world application. Performance consistency analysis across attention patterns and tasks reveals the underlying performance inconsistency across tasks for a certain efficient attention. The interpolation/extrapolation study of attention provides insight that different efficient attentions on LM/MLM behave very differently in extrapolation, which would be beneficial to future development of scalable large pre-trained models.
>
> > Q2: They claim that related codes will be released at https://github.com/Anonymous. However, I cannot find it.
>
> A2: According to "Double blind reviewing", we use an anonymous URL. In the final version, we will replace it with our github URL. We add our code in Supplementary Material.
>
> [1] Xiaoyu Shen, Ernie Chang, Hui Su, Cheng Niu, and Dietrich Klakow. Neural data-to-text generation via jointly learning the segmentation and correspondence. ACL 2020.
>
> [2] Dušek, Ondřej, Jekaterina Novikova, and Verena Rieser. Evaluating the state-of-the-art of end-to-end natural language generation: The e2e nlg challenge. Computer Speech & Language 2020.
>
> [3] Ernie Chang, Xiaoyu Shen, Dawei Zhu, Vera Demberg, and Hui Su. 2021. Neural Data-to-Text Generation with LM-based Text Augmentation. EACL 2021.
>
> [4] Ye Liu, Yao Wan, Lifang He, Hao Peng, and Philip S. Yu. Kg-bart: Knowledge graph-augmented bart for generative commonsense reasoning. AAAI 2021.
>
> [5] Hanqi Jin, Tianming Wang, and Xiaojun Wan. Semsum: Semantic dependency guided neural
> abstractive summarization. AAAI 2020.
>
> [6] Jingqing Zhang, Yao Zhao, Mohammad Saleh, and Peter Liu. Pegasus: Pre-training with extracted gap-sentences for abstractive summarization. In International Conference on Machine Learning, pp. 11328–11339. PMLR, 2020.
>
> [7] Ashish Vaswani, Noam Shazeer, Niki Parmar, Jakob Uszkoreit, Llion Jones, Aidan N Gomez, Łukasz Kaiser, and Illia Polosukhin. Attention is all you need. NeurIPS 2017.
>
> [8] Alec Radford, Jeffrey Wu, Rewon Child, David Luan, Dario Amodei, Ilya Sutskever, et al. Language models are unsupervised multitask learners. OpenAI blog 2019.
>
> [9] Tom Brown, Benjamin Mann, Nick Ryder, Melanie Subbiah, Jared D Kaplan, Prafulla Dhariwal,
> Arvind Neelakantan, Pranav Shyam, Girish Sastry, Amanda Askell, Sandhini Agarwal, Ariel
> Herbert-Voss, Gretchen Krueger, Tom Henighan, Rewon Child, Aditya Ramesh, Daniel Ziegler,
> Jeffrey Wu, Clemens Winter, Chris Hesse, Mark Chen, Eric Sigler, Mateusz Litwin, Scott Gray, Ben-
> jamin Chess, Jack Clark, Christopher Berner, Sam McCandlish, Alec Radford, Ilya Sutskever, and
> Dario Amodei. Language models are few-shot learners. NeurIPS 2020.
>
> [10] Naihan Li, Shujie Liu, Yanqing Liu, Sheng Zhao, and Ming Liu. Neural speech synthesis with transformer network. AAAI 2019.

---

> ### Author Response · Authors · 2022-12-08
> **Looking forward to receiving your feedback**
>
> Dear Reviewer MeQL,
>
> In the previous section, we elucidate the contribution of the paper and how it helps the research community.  We would like to know if our responses address your concerns.  If you have any additional questions, please feel free to discuss with us.  If you are satisfied with our response and changes, please consider reassessing our paper.
>
> Best wishes,
>
> Authors

---

### Official Review · Reviewer_kTjS · 2022-10-25

**Confidence:** 3
**Clarity, Quality, Novelty And Reproducibility:** Can you list the SOTA results with mo…
**Correctness:** 3
**Technical Novelty And Significance:** 3
**Empirical Novelty And Significance:** 3
**Recommendation:** 6

**Strength And Weaknesses:**

Strength:
1. A benchmark to evaluate different types of attention patterns for the task with long sequences
2. The benchmark covers a broad number of tasks.

Weakness:
1. The number of baselines in figures b,c,d are limited. It would be better to list more.
2. LongT5, "Efficient Text-To-Text Transformer for Long Sequences", also works on the causal attention mechanism. All the datasets in LongT5 could also be a good benchmark. Language model is also a widely used task to evaluate causal attention. Thus, the benchmark may not be novel enough if only considering the NLP area.

**Summary Of The Paper:**

 The authors propose Comprehensive Attention Benchmark (CAB) under a fine-grained attention taxonomy with four distinguishable attention patterns, namely, noncausal self, causal self, noncausal cross, and causal cross attentions. The benchmark collects seven real-world tasks from diverse fields of computer vision, natural language processing, speech processing, and time series forecasting. And an interesting finding is that some efficient Transformers often achieve less competitive results in the causal cross scenario.

**Summary Of The Review:**

The benchmark is interesting and covers many tasks from different domains. The exploration of different attention mechanism is interesting. My major concern is that there has been benchmarks for NLP tasks. After all, I would like to see people working on this new benchmark.

---

> ### Author Response · Authors · 2022-11-08
> **Response to Reviewer kTjS**
>
> Thank you for taking the time to review our paper and providing feedback. We respond to your questions below:
>
> > Q1: The limited number of baselines under causal self, noncausal cross, and causal cross attentions.
>
> A1: Most previous efficient attention work focuses on noncausal self attention pattern, which is advocated by LRA, but ignore the efficient cross/causal attention. Most of these noncausal self attentions are difficult to work on causal self, noncausal cross, and causal cross tasks, as mentioned in our paper. We will progressively include new efficient attentions in the future.
>
> > Q2: LongT5, "Efficient Text-To-Text Transformer for Long Sequences", also works on the causal attention mechanism.
>
> A2: The LongT5 model is a pretrained encoder-decoder architecture for long sequence modeling. However, LongT5 only adopts efficient attention at encoder side, while its decoder uses vanilla one. Thus we categorize LongT5's attention into noncausal self attention. Besides, the efficient attention in LongT5 consists of a local and a global one, and CAB tests a similar architecture LongShort Transformer, which is also a local-global attention and shows strong capability in causal sequence modeling.
>
> > Q3: All the datasets in LongT5 could also be a good benchmark. Language model is also a widely used task to evaluate causal attention. Thus, the benchmark may not be novel enough if only considering the NLP area.
>
> A3: The datasets in LongT5 only consider NLP domain. Attention mechanism is becoming an commonly important issue across a large variety of fields, and the development of efficient attention would be beneficial to these fields. To cover a wide range of audiences, CAB makes trade-offs when choosing tasks from diverse fields.
>
> For language tasks, we include summarization task as a straightforward real-world application facing long sequence chanllenge. We also include LM and MLM tasks in CAB for their great impact via large-scale pre-training. In the future, we will include a wider range of tasks.
>
> > Q4: Can you list the SOTA results with model pertaining for NLP tasks?
>
> A4: Pegasus [1] reports the SOTA results with model pre-training on Sum task: PEGASUS-LARGE  47.52/18.72/24.91 (R-1/R-2/R-L).  We here did not conduct experiments on model pretraining. Because an extra pretraining stage makes the evaluation of efficient attention expensive, which is unbearable for many researchers and harms the range of the benchmark audience.
>
> [1] Jingqing Zhang, Yao Zhao, Mohammad Saleh, and Peter Liu. Pegasus: Pre-training with extracted gap-sentences for abstractive summarization. In International Conference on Machine Learning, pp. 11328–11339. PMLR, 2020

---

### Official Review · Reviewer_auuz · 2022-10-27

**Confidence:** 4
**Correctness:** 3
**Technical Novelty And Significance:** 2
**Empirical Novelty And Significance:** 4
**Recommendation:** 8

**Clarity, Quality, Novelty And Reproducibility:**

***Clarity***: The paper is mostly clear and easy to read.

***Quality***: The overall quality of the submission is acceptable.

***Novelty***: The benchmark is novel, more comprehensive that the previous benchmarks and would be useful to compare future approaches and help spur research into efficient approaches.

***Reproducibility***:  Should not be a concern as the authors have promised that all the code will be released on github.


**Strength And Weaknesses:**

**Strengths**

- CAB is a novel benchmark with a finer granularity of attention taxonomy than any existing benchmark. LRA addresses only one of the 4 types of attention mechanisms.
-  CAB comprises 7 tasks of interest to the research community on 9 datasets being used by the research community across 4 domains.
- The paper also evaluates the performance of 8 popular models as baselines as a part of CAB for new approaches to be benchmarked against.
- The paper further identifies the scenarios where efficient approaches are significantly lagging behind vanilla attention in performance as the unmet need for the research community to focus on.

**Weaknesses**

***Analysis of Results***
- The paper should provide more insights on why results from certain methods vary across various studies: local attention in (Xiong et al, 2022) vs LRA; S4D on LRA vs Sum, MLM tasks, etc.
- (Figure 3) In the ‘self’ regime, why is it that performance holds across LRA and NS but drops (in correlation) for causal-self?
- (Figure 3) There is a clear blockiness along the diagonals indicating high positive correlation in performance of approaches within the ‘self’ scenarios (LRA, non-causal and causal) and within the ‘cross’ scenarios (non-causal as well as causal). What may be the reasons for high negative correlation in the off-diagonal blocks? In other words, can you kindly elaborate on footnote 4? Also, what explorations need to be done beyond this benchmark?
- It is not clear what ‘non-homologous information’ is. Kindly provide explanation and a reference for the term.

***Clarity***
- Figures showing the tradeoff between performance (accuracy) and cost (time, memory) are recommended to communicate the tradeoffs better.
- Writing should be improved in Sections 4 and 5. The authors should avoid using words/ phrases like ‘utterly defeated’, ‘embarrasingly’,  etc.
- Appropriate capitalization should be used in the ‘References’ section.
- Some easily fixable typos.


**Summary Of The Paper:**

The paper proposes a novel benchmark, called CAB,  to compare the efficiency of different attention mechanisms in transformers. The current widely used benchmark LRA focusses only on noncausal self-attention. CAB seeks to evaluate four types of attention mechanisms: the combinations of causal/ non-causal and self/ cross attention. The benchmark consists of nine datasets for seven real-world tasks across vision, NLP, speech and time-series forecasting. CAB  is also used to evaluate 8 baseline models covering the four attention mechanisms using standard statistical performance metrics and a proposed composite index as well their efficiency – relative running times and memory usage as a function of the sequence length, highlighting the strengths and weakness of efficient approach across the four attention types suggesting where more research is needed.


**Summary Of The Review:**

The paper introduces a new benchmark for comparing and evaluating efficient attention mechanisms. I expect the benchmark to be useful to the research community and is likely to help in guiding research into the design of novel efficient attention mechanisms, especially in the causal and cross-modal domains. There is an extensive benchmarking of exemplary baseline algorithms. The paper is mostly written well (though the writing, clarity and the discussions can be somewhat improved).

---

> ### Author Response · Authors · 2022-11-08
> **Response to Reviewer auuz**
>
> Thanks for your detailed comments on our paper!
>
> > Q1: Why results from certain methods vary across various studies: local attention in (Xiong et al, 2022) vs LRA; S4D on LRA vs Sum, MLM tasks, etc.
>
> A1: Xiong et al. (2022) provides an insightful observation: "With proper tuning, the results of several models could be significantly improved, (e.g., Sinkhorn, Linformer, Reformer, Performer) and there is no significant performance gap between any of the models when using a similar level of compute (measure by FLOPS)".
>
> Given the results in (Xiong et al, 2022), we guess the inconsistent performance of local attention and S4D results from under-tuned baseline models in LRA. We further provide insight on task-to-task performance in Fig 4, showing that a strong efficient attention on one task can be less competitive on another, especially across different attention patterns.
>
> > Q2: Why is it that performance holds across LRA and NS but drops (in correlation) for causal-self?
>
> A2: 1. LRA focuses on noncausal self attention evaluation with understanding tasks which is the same as the NS pattern in CAB. Fig 4 shows that tasks under the same attention pattern are more likely to positively correlate with each other in performance.
>
> 2. Modeling causality has great implications for the design of efficient attention, especially for the kernel based ones. Efficient CS attentions usually turn to a recurrent solution as cumsum operation [1] to model a partial context in order, where efficient NS attentions learn to aggregate and compress a full context of features. Modeling context in order for CS attention usually leads to a significant increase of memory and further simplification drives the causal model away from the original noncausal version [1]. Thus, efficient attentions having strong power in modeling NS pattern do not ensure high performance under the CS pattern.
>
> > Q3: What may be the reasons for high negative correlation in the off-diagonal blocks?
>
> A3: As shown in Figure 3, high negative correlation is mainly reflected between cross attention and self attention.
>   1. On the one hand, self attention adopts the query, key, and value from the same sentence which are intrinsically aligned in position, while cross attention adopt query and key/value from different sentences without any alignment information. The existence of underlying alignment information leads to the performance gap between self and cross attentions.
>   2. On the other hand, there are few works on cross attention. The small number of samples leads to statistical variance in the estimated correlations between patterns, leading to high negative correlation.
>
> > Q4: It is not clear what ‘non-homologous information’ is. Kindly provide explanation and a reference for the term.
>
> A4: We use "non-homologous information" to describe the input of cross attention. Cross attention processes the query from a target sequence and key/value from a source sequence, where the query and key/value are non-homologous. In self attention, query, key, and value are from the same sequence, where query, key, and value are homologous information.
>
> > Q5: What explorations need to be done beyond this benchmark?
>
> A5: As is suggested in Table 3, we believe there is a few exploration to be done in the future:
>   1. Efficient cross attentions and causal attentions are fairly important but are largely under-explored.
>   2. Given the performance drop in causal attention, it is also worth exploring noncausal models in both understanding and generation.
>
> [1] Hao Peng, Jungo Kasai, Nikolaos Pappas, Dani Yogatama, Zhaofeng Wu, Lingpeng Kong, Roy Schwartz, and Noah A. Smith. ABC: Attention with bounded-memory control. ACL 2022.

---

### Decision · Program_Chairs · 2023-01-20

**Decision:**

Reject

**Justification For Why Not Higher Score:**

N/A

**Justification For Why Not Lower Score:**

N/A

**Metareview: Summary, Strengths And Weaknesses:**

The paper collects a benchmark with seven real-world tasks across different domains to study attention patterns (claims to be dedicated to sequence modeling tasks).

Strength:
- The ablation study of various modes of attention, such as noncausal, cross-attention, and unidirectional (or causal) is helpful.

Weaknesses
- The performance gap for various models in many of the CAB benchmarks is not statistically significant. This means that the tasks chosen may not be the most appropriate settings to do this ablation study.
- CAB is meant to be for modeling long sequences, though there is for example super-resolution benchmark in there which I think is not the best benchmark to be used in long sequence modeling. In particular, as properly mentioned by Saharia et al. 2022: "Super-resolution works by learning to transform a standard normal distribution into an empirical data distribution through a sequence of refinement steps, resembling Langevin dynamics." This can be done by shorter tern autoregressive models as well.
- Extrapolation is abstracted away to one single scenario which basically confirms the no-free lunch principle. The study of extrapolation in sequence modeling requires benchmarks with multiple characteristics such as changes in sampling frequency input signals, irregularly sampled data, learning dynamics beyond the training distribution, distribution shifts, and so many more. A great example of testing extrapolation capability was shown for the state-space models, S4, by Gu et al. 2022, on the Speech Command recognition benchmark, where they trained on sequences of length 16k, and then tested it with sequences sampled at 8kHz frequency. Perhaps this benchmark or variations of it could be tested in CAB.
- The proposed benchmark is heavily built around attention-based models for long sequence modeling. This is a huge inductive bias that discards the opportunity of testing natural sequence modeling baselines such as state-space models [Gu et al. 2021], Fourier neural operators [Li et al. 2020], Physics informed neural networks [Lu et al. 2019], advanced RNNs such as Long Expressive Memory [Rusch et al. 2022], Closed-form Continuous-time networks [Hasani et al. 2022], and Gated RNNs with Weighted Time-Delay Feedback [Erichson et al. 2022], let alone SoTA transformer architectures such as Mega [Ma et al. 2022].

I believe a major revision of this paper is required to elaborate on the concerns raised. Thus, I do not recommend acceptance.


**Summary Of Ac-Reviewer Meeting:**

This work was not a borderline paper until the last-minute change of scores by the reviewers. Therefore, we did not have a meeting on it.